# Evaluation of the Quality and Influence of YouTube as a Source of Information on Robotic Myomectomy

**DOI:** 10.3390/jpm12111779

**Published:** 2022-10-28

**Authors:** Hye-yon Cho, Sung-ho Park

**Affiliations:** 1Department of Obstetrics and Gynecology, Hallym University Dongtan Sacred Heart Hospital, Hwaseong 18450, Korea; 2Department of Obstetrics and Gynecology, Hallym University Kangnam Sacred Heart Hospital, Seoul 07441, Korea

**Keywords:** robotic myomectomy, Da Vinci myomectomy, YouTube, information source

## Abstract

Background: We aimed to evaluate the usefulness of YouTube videos for helping patients become informed about robotic myomectomy. Methods: We searched YouTube using “robotic myomectomy” and “Da Vinci myomectomy.” Videos were sorted by view count, with the 150 most highly viewed videos being selected. From each video, content type, source, view count, video length, time on YouTube, likes, and dislikes were extracted. A scoring system was used to evaluate video quality. Results: The most prevalent content was recordings of actual surgical procedures of robotic myomectomy, and the most common provider was gynecologic surgeons. Videos directly related to robotic myomectomy were mainly provided by medical groups, had been present on YouTube for a significantly longer time (*p* = 0.003), and had a higher rate of no responses from viewers (*p* = 0.014) than videos indirectly related to robotic myomectomy. Videos uploaded by nonmedical groups had more likes, more dislikes, and a higher view ratio (*p* = 0.029, 0.042, and 0.042, respectively). Scores reflecting video quality did not differ between the two groups. Multiple logistic regression revealed that low-quality videos (less than score 5) were significantly correlated with content indirectly related to robotic myomectomy, poor general quality, fewer views, fewer likes, and no response by viewers. Conclusions: Patients who want to get informed about robotic myomectomy on YouTube should exclude low-quality videos according to such parameters as content, views, and response by viewers. In addition, medical groups should provide videos of good quality for instructing patients about this procedure.

## 1. Introduction

Robotic myomectomy has been increasingly performed for treating uterine myoma in women who want to preserve their fertility. It provides a qualified visual field with a 3D camera and wrist-like movement of instruments without tremor, which makes it easy to repair the myometrium after removing uterine fibroids. To prevent uterine rupture in a future pregnancy, primary repair of the myometrium is particularly important. Therefore, robotic myomectomy is commonly preferred for women who wish to become pregnant.

With the widespread use of the internet, patients can easily obtain health-related information from various online resources. In recent surveys, it has been shown that health-related information is commonly accessed by internet users, with 8 of 10 having accessed such information online [1]. Among the diverse resources from which such information is available online, YouTube is the most popular website, with an abundance of videos [2]. Although many physicians may agree on the importance of YouTube as a source of medical information for the public, the quality and reliability of such videos remain controversial. Given that such health-care-related videos can be uploaded without any oversight or peer review, they may contain inappropriate or misleading information.

To the best of our knowledge, no studies have evaluated the usefulness of YouTube videos for helping patients become informed about robotic myomectomy. Therefore, in this study, we evaluated the reliability and quality of the most viewed videos about this procedure uploaded to YouTube. We also analyzed the responses of viewers to determine whether viewers “liked” videos of good quality.

## 2. Materials and Methods

### 2.1. Search Strategy

A search of YouTube as described above was performed on 18 November 2021 using the term “robotic myomectomy.” Sorting of the videos in terms of their view count was performed to identify those that had been watched most frequently. Videos were included in this study if they met the following inclusion criteria: (1) in English and (2) primarily related to robotic myomectomy. They were not used in this study if they met the following exclusion criteria: (1) in languages other than English, (2) irrelevant content, (3) duplicated videos, and (4) lacked audio. Of the 300 videos that were initially identified, the top 150 were included for review.

### 2.2. Video Assessment

We obtained the following data from each video: (1) video title, (2) uploader’s name, (3) number of views, (4) length of the video (minutes), (5) uploaded date, (6) days since upload, (7) the counts of “likes” and “dislikes” (represented by the “thumbs up” and “thumbs down” icons), and (8) details of the robotic myomectomy procedure shown in the video and of the operators and associated institution.

Based on the content, we categorized the videos into four groups: (1) explanations of the surgery (providing general information about robotic myomectomy), (2) surgical procedure (detailing the techniques and processes used during the surgery, or describing the particular instruments used), (3) personal experiences (sharing personal feelings and experiences related to robotic myomectomy, e.g., pregnancy after surgery), and (4) pre- and postoperative care (e.g., bowel preparation, laboratory tests, pain control, exercise, and wound care). In addition, videos were classified into directly related (including explanations of surgery and surgical procedures) and indirectly related groups (including personal experiences or postoperative care).

According to their authorship, we classified the videos into four groups: (1) those made by academics (i.e., those affiliated with a university), (2) those made by physicians (i.e., those not affiliated with a university but working as physicians), (3) those made by patients (i.e., women who had undergone or were going to undergo robotic hysterectomy), and (4) those made by commercial entities (i.e., videos marketing a product or service). Videos were further divided into medical (uploaded by academics and physicians) and nonmedical groups (uploaded by patients and commercial entities).

### 2.3. Quality Assessment

Given the lack of established standards for evaluating the quality of videos such as those on YouTube, we used a scoring system established in a previous study [3]. For this evaluation, we employed factors related to general video quality, factors related to the inclusion of important information about robotic myomectomy and explanations of such information, and the extent to which scientific evidence was provided (Box 1). Regarding the general quality of videos and their flow, scoring was performed on a scale from 1 to 3. Meanwhile, regarding the information about robotic myomectomy, six different elements were used (indication, complication, surgical process, preoperative preparation, anesthesia, and postop management) and allocated 0 points if they were not mentioned and 2 points if they were. Regarding the provision of scientific evidence (clear statements about the sources of information and the inclusion of details about where further information about the topic of the video could be obtained) in the videos, this was assigned 0 points if such information was not provided and 3 points if it was. This gave an overall score ranging from 1 to 18 points. We determined that videos scoring less than 5 points were of low quality.

Box 1Factors for evaluating the quality of robotic myomectomy-related YouTube videos.General quality (poor: 1 point; moderate: 2 points; good: 3 points)  Overall quality (visual and audio)  Flow of contentDegree to which information in helpful to viewers (not mentioned: 0 points; mentioned: 2 points)  Indication  Complication  Surgical process  Preoperative preparation, anesthesia  Postoperative management and careScientific evidence (no: 0 points; yes: 3 points)  Clearly discloses sources of information  Provides clues of where additional information on the video topic available

The quality of each video was independently assessed by two professional gynecologists, with the average of their scores being used for analysis. The two gynecologists had performed more than 300 cases of robotic surgery (e.g., robotic myomectomy, robotic hysterectomy, and robotic ovarian cystectomy) each.

For evaluating video popularity, the like ratio (likes × 100/[likes + dislikes]), view ratio (number of views/day), and video power index (VPI) (like ratio × view ratio/100) were employed [4].

### 2.4. Statistical Analysis

In this study, SPSS Version 26.0 (SPSS, Inc., Chicago, IL, USA) and Medcalc software 15.2.2 (Medcalc, Ostend, Belgium) were used for the statistical analysis. Data are described as median (range) for continuous variables and *n* (%) for categorical ones. Cohen’s kappa coefficient was used to assess the agreement between the two reviewers, with values >0.8 representing excellent agreement, values of 0.6–0.8 representing substantial agreement, values of 0.4–0.6 representing moderate agreement, and those <0.4 representing poor agreement [5]. The ratings of the reviewers were found to be highly correlated (Cohen’s kappa coefficient 0.9).

Pearson’s chi-squared test or Fisher’s exact test was used to compare the results for categorical variables. The Kolmogorov–Smirnov test was used to assess whether the data of continuous variables were normally distributed. Variables that were normally distributed were compared using the independent *t*-test, while the Mann–Whitney U test was used otherwise. A *p*-value < 0.05 was used to indicate a statistically significant difference.

Multivariate analysis was performed using binary logistic regression. The hazard ratio (HR) and 95% confidence intervals (CI) were calculated. Cutoff values of parameters in discrimination of low-quality videos (score < 5) from others were determined by receiver-operating characteristic (ROC) curve analysis.

## 3. Results

### Main Results

Among the 150 most viewed videos related to robotic myomectomy, 121 were recordings of the actual surgical procedure, mainly performed by gynecologic surgeons. The most common source of the videos was medical groups, including physicians and academics (Table 1). Videos indirectly related to robotic myomectomy mainly involved patients sharing their experiences and feelings (Table 1). Videos sourced from commercial channels presented surgical instruments (e.g., CO_2_ laser and ultrasound) that can be used during the surgery.

The medical groups mainly contributed videos directly related to robotic myomectomy (113/120; 94.2%), whereas the nonmedical groups mainly contributed videos indirectly related to robotic myomectomy, including those on personal experiences and pre- and postoperative care (13/30; 43.3%).

Descriptive features of the videos are shown in Table 2. The top 150 videos related to robotic myomectomy had been viewed 5,445,322 times (median 919.5, range 124–4,268,240). The most viewed video had been uploaded by a medical media channel in 2014, which provides not only the detailed surgical procedure but also general information about robotic myomectomy, such as the indication for surgery, type of fibroids, and other treatment options. This video had also received the highest numbers of likes (10,000) and dislikes (1400). Among the 130 videos directly related to robotic myomectomy, 27 videos received no responses (no dislikes or likes) from viewers.

To evaluate the quality of the videos, we used a scoring method described previously [6]. To increase the accuracy of the information, we gave 3 points when scientific evidence was presented in the video. The median score was 8 (range 3.5–16). One of the three videos with the highest scores was the video with the highest numbers of views, likes, and dislikes. The other two videos were provided by physicians, which comprehensively described the surgical procedure and pre- and postoperative care.

Videos directly related to robotic myomectomy were significantly associated with a longer time on YouTube (*p* = 0.003) and more commonly had no responses from viewers (*p* = 0.014) than videos indirectly related to robotic myomectomy (Table 3). Videos uploaded by nonmedical groups had higher numbers of likes and dislikes, and a higher view ratio (*p* = 0.029, 0.042, and 0.042, respectively) (Table 4). Also, VPI tended to be higher for videos uploaded by nonmedical groups than those uploaded by medical groups (*p* = 0.062). Quality scores were similar between the two groups (Table 4).

In addition, the characteristics of the videos were compared depending on the time of upload (until 31 December 2016 vs. from 1 January 2017, onwards). The like ratio tended to be higher for videos uploaded after 2016 (*p* = 0.059). Although quality scores were higher for videos uploaded after 2016, this was not statistically significant (*p* = 0.237) (Table 5).

Multiple logistic regression revealed that low-quality videos (less than score 5) were significantly correlated with content (indirectly related to robotic myomectomy), poor general quality, views (fewer than 1472), likes (fewer than 10), and no response by viewers (Table 6).

## 4. Discussion

Since the introduction of the Da Vinci surgical system (Intuitive Surgical Inc) to gynecologic surgery in 2005, robotic myomectomy has become one of the best options for treating uterine myoma for women who want to become pregnant [7]. Robotic myomectomy can provide a safe and precise approach to the surgical field with minimal incision, as well as less pain and faster recovery.

With the trends in minimal invasive procedures, robotic myomectomy has also been increasingly attracting women with symptomatic uterine myoma.

Therefore, we reviewed the most viewed 150 videos on YouTube, and evaluated the quality and influences of each content.

The video quality scores were similar regardless of content (directly vs. indirectly related to robotic myomectomy), source (medical vs. nonmedical), and the time of upload (until 31 December 2016 vs. from 1 January 2017, onwards).

Since the videos most commonly featured recordings of the actual surgical procedure of robotic myomectomy by gynecologic surgeons, these might be less informative for members of the public interested in robotic myomectomy. In addition, some videos uploaded by surgeons had poor visual and audio quality. Moreover, most of the videos uploaded by medical groups did not provide patient-friendly information, which is valuable for those preparing to undergo robotic myomectomy.

Interestingly, no responses by viewers (no likes or dislikes) were significantly common for videos directly related to robotic myomectomy. In addition, videos uploaded by nonmedical groups were more frequently liked and disliked by viewers than those uploaded by medical groups. The rate of a lack of responses by viewers was higher for videos uploaded by medical groups than for those uploaded by nonmedical groups, although the statistical significance was marginal. Since the time on YouTube was significantly longer for videos directly related to robotic myomectomy and videos uploaded by medical groups, it is disappointing that viewers were not interested in those videos. We consider that those videos directly related to robotic myomectomy and uploaded by medical groups were not attractive to viewers because such videos contain difficult medical terms and only feature the surgical procedure, making them difficult for the general public to understand.

However, recent videos uploaded after 2016 showed slight advances compared with videos from the preceding period. The like ratio and quality score tended to be higher for recent videos, although this was not statistically significant. The rate of having no responses from viewers was lower for recent videos, although this was not statistically significant.

Most research evaluating the quality of YouTube videos about surgical procedures suggested that YouTube is an inappropriate source of public information on such procedures, which corresponds with our results [8].

Several studies have evaluated YouTube videos as a source of information about obstetric and gynecologic surgeries [9]. For example, a recent study assessed the quality of information in YouTube videos regarding hysterectomy [8]. It reported that only 6% of videos could be considered excellent, 43% moderate, and 51% poor in terms of information content [8]. In addition, most patient-made videos were critical of hysterectomy (71.72%) [8]. Meanwhile, most academic- or physician-made videos were educational and focused on the surgical techniques, and thus were aimed at doctors, not patients [8].

Another study focused on YouTube to evaluate the overall quality of the top 100 videos on the portal about cesarean delivery [6]. Among these 100 videos, 47 were directly related to cesarean delivery, and most (*n* = 30) of the videos had been produced by physicians [6]. It was also found that most videos directly related to cesarean delivery had been produced by medical groups, which received higher quality scores than the videos indirectly related to such delivery mainly provided by nonmedical groups [6]. The results also revealed that the videos directly related to cesarean delivery had often been provided at an earlier date and had lower like ratios than the videos indirectly associated to cesarean delivery [6].

This study is the first to analyze the quality and influence of YouTube videos about robotic myomectomy. However, it has some limitations. First, we analyzed only videos using the English language, which may have caused selection bias. In future study, it will be necessary to include videos using other languages to achieve a more comprehensive evaluation. Second, we chose to use a quality scoring system referred to in previous reports, as there are no validated criteria for video analysis. We added essential information on robotic myomectomy for the quality evaluation of videos. In addition, to increase the accuracy of the information, we gave 3 points when scientific evidence was presented in the video. However, there is a need for more validated criteria to comprehensively evaluate video quality.

## 5. Conclusions

In conclusion, YouTube is not yet a valuable source of adequate information about robotic myomectomy. Although medical groups have long uploaded many videos directly related to this procedure, most of these videos just feature recordings of the procedure in real time, thus providing no easily understandable information for the general public seeking health-care information on YouTube. Therefore, medical groups who are responsible for general health care should provide videos of good quality for patients.

In addition, videos indirectly related to robotic myomectomy with poor general quality, fewer views (fewer than 1472), fewer likes (fewer than 10), and no response by viewers are not recommended for patients who want to get informed about robotic myomectomy on YouTube.

## Figures and Tables

**Table 1 jpm-12-01779-t001:** Characteristics of robotic myomectomy-related YouTube videos (N = 150).

Variable	Description	Value, *n*
Content		
Directly related		130
	Explanations of surgery (provides general robotic myomectomy-related information) or surgical procedure (shows or explains surgical techniques and processes in detail)	
Indirectly related		20
	Personal experiences (shares personal experiences and feelings related to robotic myomectomy) or on postoperative care (provides information about postoperative care, e.g., bowel preparation, laboratory test, pain control, exercise, and wound care)	
Source		
Medical		120
Academic	Authors are affiliated with a university	36
Physician	Authors are physicians but not affiliated with a university	84
Nonmedical		30
Patient	Woman who had undergone or was expecting to undergo robotic hysterectomy	10
Commercial	Attention to a service or product	20

**Table 2 jpm-12-01779-t002:** Descriptive features of robotic myomectomy-related YouTube videos (*n* = 150).

Variable	Median (Range)
Views, *n*	919.5 (124–4,268,240)
Video length (minutes)	6 (1–148)
Time on YouTube (days)	2671 (101–5238)
Likes, n	5 (0–10,000)
Dislikes, n	0 (0–1400)
Like ratio	100 (60–100)
View ratio	0.497 (0.03–1598.59)
Video power index	0.578 (0.05–1402.27)
Score	8 (3.5–16)

**Table 3 jpm-12-01779-t003:** Content comparison of the videos.

Variable	Mean ± SD, n (%)		*p* Value
	Directly Related (*n* = 130)	Indirectly Related (*n* = 20)	
Views, *n*	41,498.4 ± 375,154.79	2526.4 ± 4054.89	0.239
Video length (minutes)	9.5 ± 17.84	9.80 ± 8.19	0.889
Time on YouTube (days)	2666.9 ± 1183.53	1818.9 ± 1065.42	0.003 *
Likes, n	95.9 ± 876.96	36.6 ± 45.54	0.445
Dislikes, n	12.2 ± 122.77	1.1 ± 2.70	0.304
Like ratio	96.3 ± 7.79	37.9 ± 5.68	0.285
View ratio	15.0 ± 140.21	1.7 ± 2.62	0.283
Video power index	16.7 ± 138.11	1.7 ± 2.52	0.274
Score	8.2 ± 2.79	6.9 ± 3.38	0.117
No response by viewers	27 (20.8)	0 (0.0)	0.014 *

* *p* Value < 0.05.

**Table 4 jpm-12-01779-t004:** Source comparison of the videos.

Variable	Mean ± SD, *n* (%)		*p* Value
	Medical Group (*n* = 120)	Nonmedical Group (*n* = 30)	
Views, *n*	8494.2 ± 35,391.83	147,533.9 ± 778,354.08	0.051
Video length (minutes)	9.9 ± 18.50	8.1 ± 7.28	0.404
Days on YouTube	2681.5 ± 1201.13	2043.3 ± 1071.13	0.006
Likes, *n*	15.6 ± 42.00	377.8 ± 1818.97	0.029 *
Dislikes, *n*	1.2 ± 5.05	48.6 ± 255.28	0.042 *
Like ratio	96.8 ± 7.64	95.6 ± 7.02	0.413
View ratio	2.4 ± 7.93	56.6 ± 291.32	0.042 *
Video power index	2.7 ± 7.82	53.4 ± 264.43	0.062
Score	8.2 ± 2.84	7.5 ± 3.12	0.281
No response by viewers	25 (20.8)	2 (6.7)	0.054

* *p* Value < 0.05.

**Table 5 jpm-12-01779-t005:** Comparison of the time of video upload to YouTube.

Variable	Mean ± SD, *n* (%)		*p* Value
	Until 31 December 2016 (*n* = 109)	From 1 January 2017, Onwards (*n* = 41)	
Views, *n*	48,632.2 ± 409,583.45	3522.4 ± 9602.25	0.253
Video length (minutes)	9.4 ± 19.05	9.9 ± 8.85	0.824
Time on YouTube (days)	3133.2 ± 802.20	1013.8 ± 533.90	<0.0001 *
Likes, *n*	108.1 ± 957.25	34.6 ± 68.13	0.427
Dislikes, *n*	14.2 ± 134.07	1.3 ± 3.75	0.315
Like ratio	95.7 ± 8.46	98.5 ± 3.76	0.059
View ratio	17.2 ± 153.10	2.9 ± 5.94	0.333
Video power index	18.9 ± 150.26	3.1 ± 5.96	0.331
Score	7.9 ± 2.95	8.3 ± 2.78	0.455
No response by viewers	22 (20.2)	5 (12.2)	0.186

* *p* Value < 0.05.

**Table 6 jpm-12-01779-t006:** Correlating factors of low-quality videos (score < 5 points).

Variable	Odds Ratio (95% Confidence Interval)	*p* Value
Content (indirectly related)	11.132 (1.917~64.655)	0.007 *
General quality (poor)	43.422 (7.687~245.275)	<0.0001 *
Views, *n* (<1472)	5.051 (1.021~24.988)	0.047 *
Likes, *n* (<10)	12.721 (1.948~83.064)	0.008 *
Time on YouTube (>6.73 years)	4.429 (0.993~19.764)	0.051
No response by viewers	35.558 (4.543~278.891)	0.001 *

* *p* Value < 0.05.

## Data Availability

Not applicable.

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
