# Peer review of "Evaluation of the Quality and Influence of YouTube as a Source of Information on Robotic Myomectomy"

_jpm, 2022, doi:10.3390/jpm12111779_

Round 1

Reviewer 1 Report

This is a very interesting investigation. Authors should review the following: Line 153 says: Table 1. Factors for evaluating the quality of robotic myomectomy-related YouTube videos. Line 154 says: Table 1. Characteristics of robotic myomectomy-related YouTube videos (N=150). The tables have the number 1, is that correct? The manuscript has no other doubts or corrections.

Reviewer 2 Report

Nice work and presentation. Interesting article in our days where internet is flooded with unvaluable information. 

Author Response

Dear the reviewer,

As you say, our subject is interesting in our days where internet is flooded with unvaluable information.

Thank you for your nice review and kind comments.

Sincerely, Dr. Cho.

Reviewer 3 Report

Study of the YouTube sources for learning robotic myomectomy is a relevant and exciting topic to the field of the journal. The text is clear and easy to read. The manuscript has an excellent methodical description. The overall paper is organized and well written. The methods, the overall study design, and statistical analysis are clearly described. I appreciate that the results achieve the proposed objectives. The discussions section is well organized, insightful, and informative. The figures and the tables are well presented and easy to read and understand. The conclusions are clear and supported by the results. The references are up to date and appropriate for the theme. Congratulations to the research team on this success.

Author Response

Dear reviewer,

Thank you for your detailed review and kind comments.

Best regards, Dr. Cho

Reviewer 4 Report

This paper addresses the quality of videos about robotic myomectomy on Youtube that is rarely mentioned. The authors reported that the videos directly related to robotic myomectomy were mainly provided by medical groups that had been present on YouTube for a significantly longer time, and had a higher rate of no responses from viewers than videos indirectly related to robotic myomectomy. Moreover, they showed that videos uploaded by nonmedical groups had more likes, more dislikes, and a higher view ratio. They stated that the quality of the videos was similar regardless of content, source and upload date. They concluded that Youtube is not a valuable source of adequate information about robotic myomectomy. I believe that this theme is novel and interesting to focus on, but that it is not well discussed. While data were collected appropriately and I appreciate that, I have serious concerns about the interpretation about that.

Major concerns:

1.       The authors evaluate videos on Youtube from several perspectives, but lack discussion to reach the final conclusion. Although they aim “to evaluate the quality and influence of Youtube videos related to robotic myomectomy”, it would be easier to discuss the evaluation if the purpose of the evaluation were narrowed down (For example, evaluating the usefulness of the videos for helping robotic surgery beginners learn the technique or for helping patients become informed about robotic surgery, etc).

2.       It would be better if suggestions as to what parameters or keywords are associated with the higher scored videos (i.e., how readers can find useful videos on Youtube) were provided in the paper. A multivariate analysis would be needed to assess the interrelationship between the scores and some parameters.

Minor concerns:

1. Are the two gynecologists who scored proficient in robotic surgery or are they beginners? That could make a big difference in the score and should be clearly stated.

Round 2

Reviewer 4 Report

The authors have responded to the comments and revised the paper. Some have been adequately addressed, but some are still inadequate.

I still have a concern about publication. I hope that my comment is useful for the improvement of the article.

Major concern:

The authors revised that the aim of this study was to evaluate the usefulness of Youtube videos for helping patients become informed about robot myomectomy, and the conclusion is that Youtube videos are not valuable sources. However, this conclusion cannot be drawn from the results presented. They need to show results that support their claim. A new parameter to measure how well patients understand about the robot myomectomy is need to be added, or if existing data indicate the degree to which patients understand the surgery, this needs to be clearly stated.

Author Response

Dear reviewer,

Thank you for your detailed review again.

According to your comment, I edited results and conclusion of Abstract.

The revised part is colored in yellow.

I wish you receive your positive response.

Thank you for your help.

Sincerely, Dr. Cho